# The Fabrication of Calcium Alginate Beads as a Green Sorbent for Selective Recovery of Cu(II) from Metal Mixtures

**Niannian Yang [1], Runkai Wang [1],\***  **, Pinhua Rao [1],\*, Lili Yan [1], Wenqi Zhang [2], Jincheng Wang [1] and Fei Chai [1]**

[1]    School of Chemistry and Chemical Engineering, Shanghai University of Engineering Science, Shanghai 201620, China; m040116128@sues.edu.cn (N.Y.); liliyan@sues.edu.cn (L.Y.); wjc406@sues.edu.cn (J.W.); m040117127@sues.edu.cn (F.C.)

[2]    School of Civil Engineering, Kashgar University, Xinjiang 844000, China; zhangwenqi@sues.edu.cn

\*    Correspondence: wangrunkai@sues.edu.cn (R.W.); raopinhua@sues.edu.cn (P.R.); Tel.: +86-021-67791217  (R.W.); +86-021-67791211 (P.R.)

**Abstract:** Calcium alginate (CA) beads as a green sorbent were easily fabricated in this study using sodium alginate crosslinking with $CaCl_2$, and the crosslinking pathway was the exchange between the sodium ion of $\alpha$-L-guluronic acid and Ca(II). The experimental study was conducted on Cu(II), Cd(II), Ni(II) and Zn(II) as the model heavy metals and the concentration was determined by inductively coupled plasma optical emission spectrometry (ICP-OES). The characterization and sorption behavior of the CA beads were analyzed in detail via using scanning electron microscopy (SEM), fourier transform infrared spectroscopy (FTIR) and X-ray photoelectron spectroscopy (XPS). The adsorption experiments demonstrated that the CA beads exhibited a high removal efficiency for the selective adsorption of Cu(II) from the tetra metallic mixture solution and an excellent adsorption capacity of the heavy metals separately. According to the isotherm studies, the maximum uptake of Cu(II) could reach 107.53 mg/g, which was significantly higher than the other three heavy metal ions in the tetra metallic mixture solution. Additionally, after five cycles of adsorption and desorption, the uptake rate of Cu(II) on CA beads was maintained at 92%. According to the properties mentioned above, this material was assumed to be applied to reduce heavy metal pollution or recover valuable metals from waste water.

**Keywords:** alginate beads; green sorbent; selective adsorption; heavy metals

## 1. Introduction

In recent years, agriculture and the mining, chemical fertilizer, leather, battery and paper industries have developed vigorously, and the phenomenon of heavy metal wastewater directly or indirectly being discharged into the environment has become more and more serious, particularly in developing countries [1,2]. Heavy metal ions in wastewater are mainly comprised of zinc (Zn), nickel (Ni), cadmium (Cd), copper (Cu) and so on. Among them, copper is widely used in industrial production, such as in printed circuit boards (PCBs), and it is also one of the indispensable nutrients (trace elements) in the human body [3,4]. When copper-containing wastewater is discharged into the environment beyond its self-purification range, the high toxicity and non-biodegradability of copper ions poses a serious threat to animal and human health. Hence, investigating how to effectively remove copper ions from wastewater is very important to the ecological environment. In addition, the recovery of copper from wastewater also has certain economic benefits.

To make the recovery of Cu more meaningful, the literature reported many methods. For example, Coruh et al. [5] selected vitrification as the technology to deal with industrial copper, mixing it with other inorganic wastes and materials and sintering them into glass for reuse. Printed circuit boards (PCBs) are widely used in the electronics industry [6,7], resulting in a large amount of copper-containing wastewater. Liu et al. [8] used electrolysis to recover 97% copper from waste PCBs, and Mdlovu et al. [9] reported the use of the microemulsion process to recover copper nanoparticles with diameters of 20–50 nm. In addition to the methods reported above, adsorption was also selected to treat copper-containing wastewater due to its advantages, such as low initial cost and process simplicity [10,11]. However, the instability of the adsorbents and difficulty in the separation still limit their practical applications. Overall, for the treatment of heavy metals in wastewater, availability and cost effectiveness play an important role in the synthesis of the adsorbents. This has made people pay attention to abundant, renewable and environmentally-friendly marine resources, such as bio-sorbents [12]. Torres-Caban et al. [13] proved that a calcium alginate/spent coffee grounds composite bead was an effective biological absorbent for the removal of copper ions, which first made us notice the role of alginate in the process of adsorption.

Alginate derived from brown algae is a highly popular material for the biosorption of heavy metals due to its advantages such as low cost and high affinity via gelation [14,15]. Abundant functional groups have been found in sodium alginate, such as carboxyl and hydroxyl groups, which can crosslink with cations [16,17]. The reason for this is that the carboxyl group is a negative group, and it can adsorb electrostatically with heavy metal ions and produce chelation at the same time. Sodium alginate reacts with divalent cations such as Ca(II), Ba(II) and Sr(II), to form insoluble hydrogels, which are crosslinked to form a reticular structure called the "egg box" structure, and the crosslinking pathway is the exchange between the sodium ions of $\alpha$-L-guluronic acid and divalent ions [18,19]. In all types of alginate materials, alginate beads can be easily recovered from water [20,21]. Consequently, calcium alginate (CA) is a promising biomaterial for the biosorption of heavy metals [22]. In previous studies, it has been found that CA has a selective adsorption effect on some metal ions, which is extremely important for the recovery and re-utilization of metal ions. Hence, we want to investigate whether CA has selective adsorption on copper under the interference of cadmium, zinc and nickel metal ions.

In our study, the alginate beads as a green sorbent were easily fabricated with the Ca(II) crosslink, maintaining the high efficiency of the selective recovery of Cu(II) from the metal mixtures and the good adsorption of the heavy metals separately. The experimental study was conducted on Cu(II), Cd(II), Ni(II) and Zn(II) as the model heavy metals. The morphology and structure, functional groups, adsorption mechanism of CA beads were investigated by various characterization methods, such as scanning electron microscopy (SEM), fourier transform infrared spectroscopy (FTIR) and X-ray photoelectron spectroscopy (XPS). All pH measurements were adopted using a LEICI PHS-2F pH meter. Additionally, the alginate beads showed good reusability after five rounds of simple sorption–desorption procedures.

## 2. Materials and Methods

### 2.1. Materials

Sodium alginate and calcium(II) chloride ($CaCl_2$) were purchased from Adamas-beta. Calcium chloride was used for crosslinking of the alginate beads and sodium alginic acid was used to fabricate the alginate beads. Four types of heavy metals, $Cu(NO_3)_2 \cdot 3H_2O$, $Zn(NO_3)_2 \cdot 6H_2O$, $Ni(NO_3)_2 \cdot 6H_2O$ and $Cd(NO_3)_2 \cdot 4H_2O$, were purchased from Aladdin. To adjust the pH of the solution, 1 M hydrochloric acid (HCl) and 1 M sodium hydroxide (NaOH) were applied, which were acquired from Sinopharm Chemical Reagent Co., Ltd. All other chemicals used in this study were of analytical grade without purification. Distilled water (DW) with specific resistivity greater than 18 M$\Omega$·cm was used in all experiments.

## 2.2. Preparation of Calcium Alginate Beads

Sodium alginate powder was added to distilled water to obtain a yellow viscous sodium alginate solution. The obtained solution remained stationary until there were no air bubbles. To synthesize the spherical bio-sorbent, sodium alginate solution was added dropwise into $CaCl_2$ (1%, w/v) solution under gentle stirring with a dropper and the bead was formed immediately. After 2 hours of curing, the sphere became compact, and a CA bead with a diameter of about 3 mm was obtained. The gel ball was rinsed with distilled water several times to remove free Ca(II) and stored in distilled water for further use.

## 2.3. Material Characterizations

The concentration of Cu(II), Zn(II), Ni(II) and Cd(II) used in all experiments was determined by inductively coupled plasma optical emission spectrometry (ICP-OES, Varian 700-ES, Walnut Creek, CA, USA), using 2% $HNO_3$ as the medium. The standard solution and the solution of heavy metal ions to be measured were acidic. Scanning electron microscopy (SEM, Hitachi SU8010, Ibaraki, Japan) was used to obtain the information of the physical structure and morphology of alginate beads. Fourier transform infrared spectroscopy (FTIR, PerkinElmer Spectrum Two, Waltham, MA, USA) was recorded in the 400–4000 $cm^{-1}$ region on a FTIR spectrophotometer using a KBr disk method. Thermogravimetric analysis (TGA, TA Q600 SDT, New Castle, DE, USA) was carried out in a nitrogen gas flow from room temperature to 600 °C with a heating rate of 10 K/min. The point of zero charge (STARTER 2100, Parsippany, NJ, USA) was measured within the pH range from 3.0 to 10.0 by addition of 0.1 N HCl and NaOH. The samples were further analyzed by X-ray photoelectron spectroscopy (XPS, Thermo Fisher 250XI, Waltham, MA, USA) for the Cu 2p, Cd 3d, Zn 2p and Ni 2p regions. The charging shifts of the spectra were calibrated by placing the C1s peak at 284.8 eV from the adventitious carbon. Then, the results obtained from XPS were analyzed using the non-linear least squares curve fitting program (XPSPEAK4.1, software, Hong Kong, China).

## 2.4. Adsorption Experiments

For kinetic studies, an adsorption experiment was carried out using 200 mg/L of Cu(II), Zn(II), Ni(II) and Cd(II) in distilled water, respectively, of which the initial pH was 5.5, 6.3, 6.6 and 6.6, respectively. The saturated adsorption time was determined, and the samples were taken at a predetermined time interval. The adsorption isotherms of CA beads in a monomer solution and a tetra-metallic mixture solution were studied to evaluate their saturated adsorption capacity. The beads (1 g) were fully dispersed in 50 mL of monometallic solution and tetra-metallic mixture solution with different concentrations ranging from 50–800 mg/L at 120 r/min for 12 h at a temperature of 25 °C in a thermostabilized warm bath. In the pH effect experiment, five groups of tetra-metallic mixture solutions with a pH of 2.0, 3.0, 4.0, 5.0 and 6.0, respectively, were prepared and adjusted by adding NaOH (0.1 mol/L) or $HNO_3$ (0.1 mol/L).

The suspended impurities in the solution were removed by membrane filtration. The concentration of the metal ions was determined by ICP-OES and the amount of adsorption could be calculated using the following equation:

$$q = \frac{(C_0 - C)V}{M} \tag{1}$$

where q is the adsorption of the metal ions (mg/g), $C_0$ and C are the initial and final metal ion concentrations, respectively (mg/L), V is the total volume of suspension (L), and M is the dry mass of adsorbent (g).

## 2.5. Desorption and Reuse Experiments

Desorption experiments were also conducted using 1% $CaCl_2$ solutions and 0.1 M $HNO_3$ as the desorption reagents. In this experiment, 1 g of CA beads were added to 50 mL tetra-metallic mixture

solutions with pH 5.7. After sorption, the heavy metal loaded CA beads were subsequently suspended in a 0.1 M $HNO_3$ eluting agent to evaluate the desorption performance. Then, the CA beads were washed with 1% $CaCl_2$ solutions to make them neutral and distilled water several times to remove the free Ca(II) ions from the beads. Four sorption–desorption cycles were conducted to assess the reusability of CA beads.

## 3. Results and Discussion

### 3.1. Characterization of CA Beads

The CA beads were synthesized through crosslinking with a diameter of 3 mm. Figure 1a shows the overall appearance of CA beads after drying; the diameter was reduced to about 1 mm. Figure 2b,c shows that there were a lot of ravines on the surface of CA beads, which increased the surface area and provided more adsorption sites. The surface of the CA bead was composed of wire-like Ca-alginate and a honeycomb network could be observed in the internal structure of CA (Figure 1d).

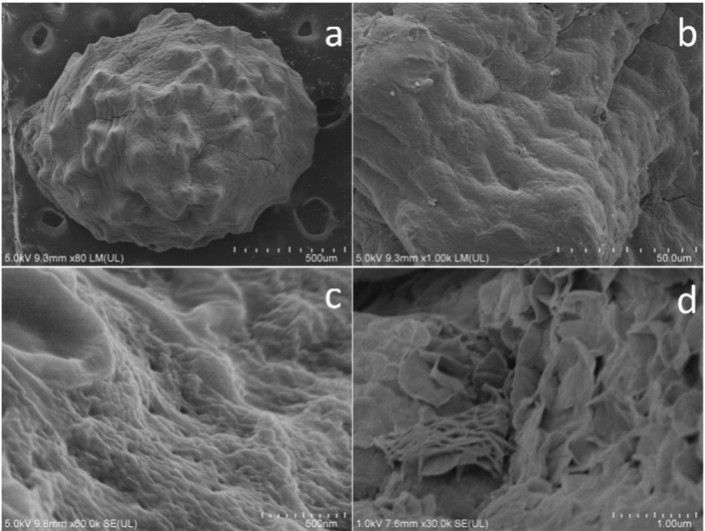

**Figure 1.** Scanning electron microscopy (SEM) images of the outer surface (**a**–**c**) and the internal structure (**d**) of the dried calcium alginate (CA) bead.

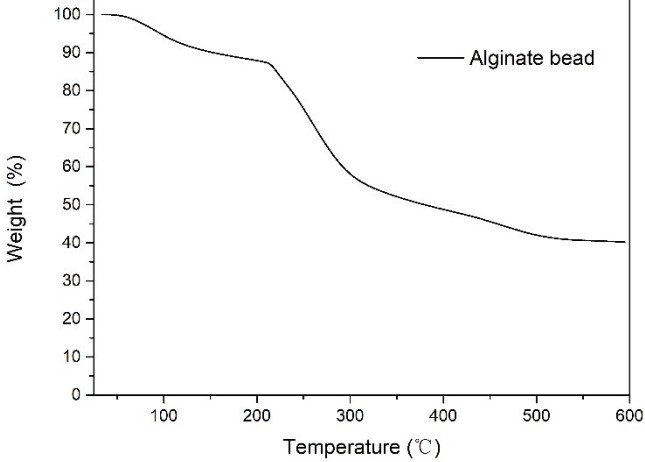

**Figure 2.** Thermogravimetric analysis (TGA) curves of the CA beads.

To evaluate the hydration and thermal decomposition of the CA beads, thermogravimetric analysis (TGA) was applied. As seen in Figure 2, the first mass loss was attributed to the dehydration process and a 12.6 wt% loss up to 210 °C. The second part was a thermal degradation stage, which resulted

in pyrolysis from 210 °C to 510 °C with a weight loss of 45.8 wt%. The third part represented the conversion of the remaining materials to carbon residues and CaCO$_3$ and the residual weights were 40.2 wt% [23].

### 3.2. Adsorption Kinetics

In industrial applications, adsorption kinetics are important for process design and operation, which required us to achieve adsorption equilibrium under certain system conditions. The kinetic behavior of the metal ions removed by CA beads is presented in Figure 3. The experimental data were fitted with pseudo-first-order and pseudo-second-order kinetics [24,25].

$$\text{Pseudo} - \text{first} - \text{order model}: \ q_t = q_1(1 - \exp(-k_1 t)) \tag{2}$$

$$\text{Pseudo} - \text{second} - \text{order model}: \ q_t = \frac{q_2^2 k_2 t}{1 + q_2 k_2 t} \tag{3}$$

where both $q_1$ and $q_2$ are the amount of metal ions adsorbed at equilibrium (mg/g), $q_t$ is the amount of metal ions adsorbed at any time t (mg/g), $k_1$ and $k_2$ are equilibrium rate constants of the pseudo-first-order and pseudo-second-order, respectively (g/mg min).

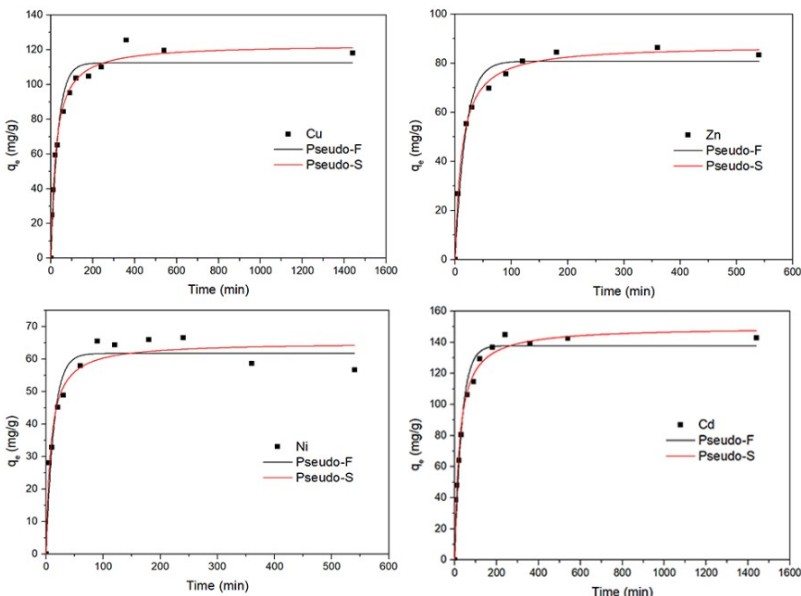

**Figure 3.** Kinetic adsorption plots of Cu(II), Zn(II), Ni(II) and Cd(II) by CA beads, respectively. Reaction conditions: T = 293 K, [Cu(II)]$_0$ = [Zn(II)]$_0$ = [Ni(II)]$_0$ = [Cd(II)]$_0$ = 200 mg/L.

The results for the kinetic model parameters are summarized in Table 1. The higher coefficient of determination (R$^2$) value of the pseudo-second-order model demonstrated that the adsorption rate was dominated by the rate of the chemical reaction [26]. In this experiment, the ion exchange occurred between the metal ions and Ca(II) and the adsorption rate of the CA was dependent on the exchange rate. The obtained k$_2$ values for Cu(II), Zn(II), Ni(II) and Cd(II) adsorption onto CA beads indicated that the adsorption rate was Ni(II) > Zn(II) > Cd(II) ≈ Cu(II).

**Table 1.** Parameters of pseudo-first-order and pseudo-second-order kinetic models. $q_1$ and $q_2$ = the amount of metal ions adsorbed at equilibrium; $k_1$ and $k_2$ = equilibrium rate constants of the pseudo-first-order and pseudo-second-order, $R^2$ = coefficient of determination, respectively.

| Metal Type | Pseudo-First-Order | | | Pseudo-Second-Order | | |
|---|---|---|---|---|---|---|
| | $q_1$ (mg/g) | $k_1$ (min$^{-1}$) | $R^2$ | $q_2$ (mg/g) | $k_2$ (g/mg min) | $R^2$ |
| Cd | 137.68 | 0.0300 | 0.9636 | 149.62 | 0.0436 | 0.9866 |
| Cu | 112.34 | 0.0304 | 0.9547 | 123.03 | 0.0422 | 0.9888 |
| Zn | 80.81 | 0.0548 | 0.9688 | 87.34 | 0.0837 | 0.9955 |
| Ni | 61.75 | 0.0743 | 0.9426 | 65.16 | 0.1243 | 0.9559 |

*3.3. Adsorption Isotherm*

In order to evaluate the maximum adsorption capacity of CA beads, the adsorption isotherms were measured in the initial metal concentrations ranging from 0–800 mg/L. The adsorption equilibrium data were fitted with Langmuir and Freundlich isotherm models [27,28].

$$\text{Langmuir model}: q_e = \frac{q_m K_L C_e}{1 + K_L C_e} \tag{4}$$

$$\text{Freundlich model}: q_e = K_F C_e^{1/n} \tag{5}$$

where $q_e$ is the equilibrium adsorption capacity of heavy metal ions (mg/g), $q_m$ indicates the maximum adsorption capacity for heavy metal ions (mg/g), $C_e$ is the equilibrium concentration after adsorption (mg/L), $K_L$ and $K_F$ denote equilibrium constants of Langmuir (L/mg) and Freundlich (mg/g) (L/g), respectively, and n is the Freundlich exponent.

Figure 4 shows the adsorption isotherm of Cu(II), Zn(II), Ni(II) and Cd(II) separately on the CA beads and Table 2 lists the parameter values along with their correlation coefficients. The CA beads showed good adsorption towards Cu(II), Zn(II), Ni(II) and Cd(II), with a maximum adsorption of 140.55 mg/g, 174.60 mg/g, 114.69 mg/g and 216.82 mg/g, respectively. As shown in Table 3, calcium alginate beads exhibited significant advantages over other low-cost biosorption materials in terms of their maximum adsorption capacity for Cu(II).

Figure 5 shows the selective removal of Cu(II) on CA beads. With the increase of equilibrium concentration, the adsorption effect of CA beads on Cu(II) first increased rapidly and then tended to slow until equilibrium was reached, while the adsorption capacity of Zn(II), Ni(II) and Cd(II) decreased slowly. The fitting parameters of the Langmuir and Freundlich models are shown in Table 2 and the adsorption data were more consistent with the Langmuir model ($R^2$ = 0.9920) than the Freundlich model ($R^2$ = 0.8126) according to the Cu(II)* adsorption coefficient, which suggested that the adsorption of Cu(II) was the mono-layer sorption during the sorption process. Due to the competitive sorption between Cu(II) and other metals, the isotherm data of Zn(II), Ni(II) and Cd(II) were unable to be fit by the Langmuir or Freundlich models. Therefore, the CA beads showed better sorption capacity toward Cu(II) selectively than the other three heavy metals, indicating that the CA beads could be applied to the selective recovery of Cu(II) from polymetallic solutions.

**Table 2.** Parameters of the Langmuir and Freundlich models. $q_m$ = the maximum adsorption capacity for heavy metal ions; $K_L$ and $K_F$ = equilibrium constants of Langmuir and Freundlich, respectively; n = Freundlich exponent; $R^2$ = coefficient of determination.

| Metal | Langmuir Model | | | Freundlich Model | | |
|---|---|---|---|---|---|---|
| | $q_m$ (mg/g) | $K_L$ (L/mg) | $R^2$ | $K_F$ (mg/g) (L/g)$^{1/n}$ | n | $R^2$ |
| Cu | 140.55 | 0.0553 | 0.9605 | 46.9337 | 5.6609 | 0.8729 |
| Cd | 216.82 | 0.0177 | 0.8917 | 31.2836 | 3.3430 | 0.8939 |
| Zn | 174.60 | 0.0055 | 0.8821 | 10.0481 | 2.4589 | 0.8570 |
| Ni | 114.69 | 0.0108 | 0.9842 | 14.1111 | 3.2582 | 0.8690 |
| Cu * | 107.53 | 0.0639 | 0.9920 | 44.6145 | 7.2228 | 0.8126 |

* Parameters of Langmuir and Freundlich models for selective adsorption of Cu(II) by CA in mixed solution.

**Table 3.** Comparison of adsorption capacity of low-cost adsorbents for Cu(II).

| Adsorbents | Maximum Adsorption Capacity of Cu(II) | References |
|---|---|---|
| IDA-modified cellulose | 69.6 mg/g | [29] |
| Alkali leaching wire rope sludge | 36.48 mg/g | [30] |
| EDTA-functionalized bamboo activated carbon | 42.19 mg/g | [31] |
| Waste coffee | 92.78 mg/g | [32] |
| Kapok-DTPA | 101.0 mg/g | [33] |
| CA beads | 140.55 mg/g | This study |

Where, IDA, EDTA and DTPA represent iminodiacetic acid, ethylene diamine tetraacetic acid, and diethylenetriamine pentaacetic acid, respectively.

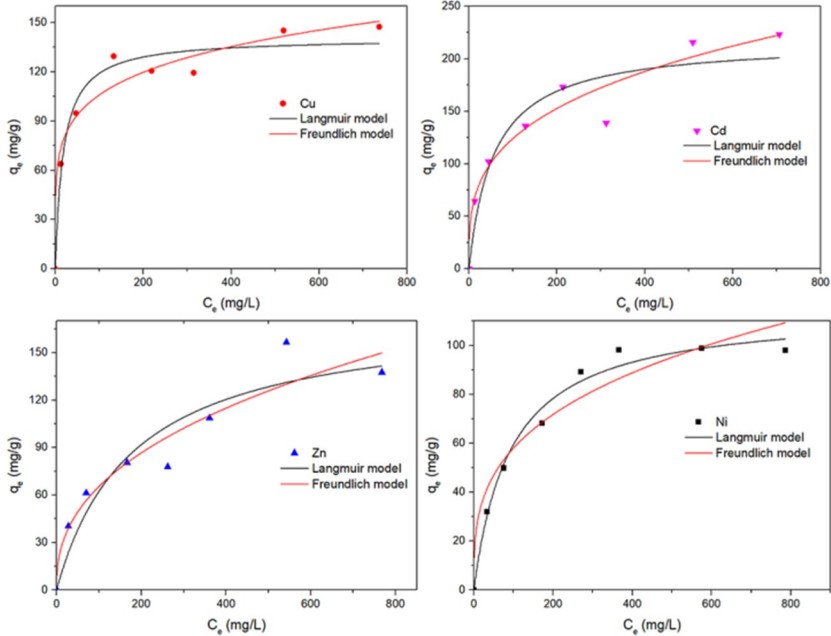

**Figure 4.** The adsorption isotherm by CA beads in the Cu(II), Zn(II), Ni(II) and Cd(II) heavy metal solution at different initial concentrations. Reaction conditions: T = 293 K, $[Cu(II)]_0$ = $[Zn(II)]_0$ = $[Ni(II)]_0$ = $[Cd(II)]_0$ = 0–800 mg/L, reaction time = 12 h.

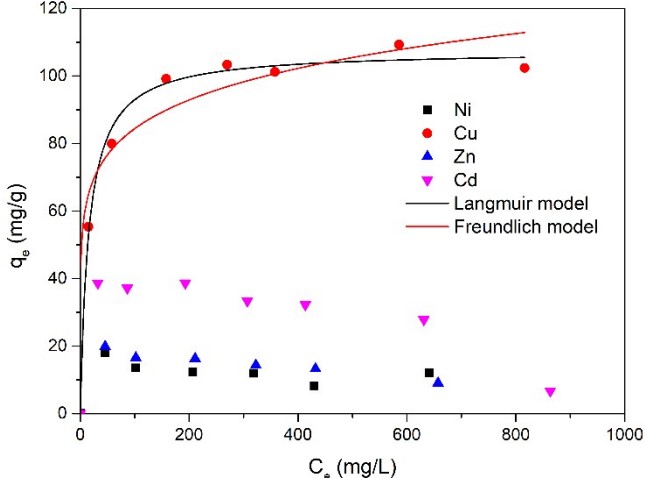

**Figure 5.** The selective removal and adsorption isotherm by CA beads in the mixed heavy metals solution at different initial concentrations. Reaction conditions: T = 293 K, $[Cu(II)]_0 = [Zn(II)]_0 = [Ni(II)]_0 = [Cd(II)]_0 = 0–800$ mg/L, reaction time = 12 h. $q_e$ = the equilibrium adsorption capacity of heavy metal ions; $C_e$ = the equilibrium concentration after adsorption.

A dimensionless $R_L$ constant was introduced to reveal the essential characteristics of the Langmuir model, and the adsorption process was further evaluated. The correlation values of $R_L$ could be calculated from the following equation.

$$R_L = \frac{1}{1 + bC_i} \tag{6}$$

where $C_i$ is the initial concentration of Cu(II) (mg/L) and b is the dimensionless constant of Langmuir. The values of $R_L$ indicated that the adsorption process may be an unfavorable trend ($R_L > 1$), linear ($R_L = 1$), favorable ($0 < R_L < 1$) or irreversible ($R_L = 0$) [34]. Here, the adsorption process of Cu(II) by CA beads showed that the b value was 0.0639, the calculated $R_L$ value was within the range of 0.0175–0.2434 and just fell within the range of 0–1, indicating that CA beads were favorable to the Cu(II) adsorption process.

*3.4. Effect of pH*

The effect of pH on Cu(II) adsorption by CA beads in the mixed metal solutions was investigated with the pH value in the range of 2–6. Figure 6 shows the uptake capacities of heavy metals using the CA beads at various pH values. When the pH value was lower than 2.0, few heavy metal ions were adsorbed due to the competition of the many $H^+$ ions. With the increase of initial pH value from 2 to 6, the adsorption capacity of CA beads to Cu(II) increased significantly, and when the pH was 6, the maximum adsorption was observed. However, there was no obvious change in the adsorption amount of the other three kinds of metal ions, which were only slightly increased.

The point of zero charge (PZC) was employed to analyze the surface charges of the CA beads, and the measured value was 8.2. When the pH was lower than 8.2, the surface of the material was positively charged because of the introduction of calcium ions. At a pH above 8.2, there was a negative charge due to the reaction between the -OH on the surface and the OH· in the solution, leading to the formation of a negatively charged functional group, $O^-$. Therefore, the adsorption process of heavy metal ions by CA beads could be affected by pH, and there is a competition mechanism between $H^+$ and metal ions with carboxyl groups in acidic environments. Additionally, the carboxylic (-COOH) and hydroxyl (-OH) groups of the CA beads were changed to the protonated forms and the Ca(II) was released into the solution under acidic conditions [35].

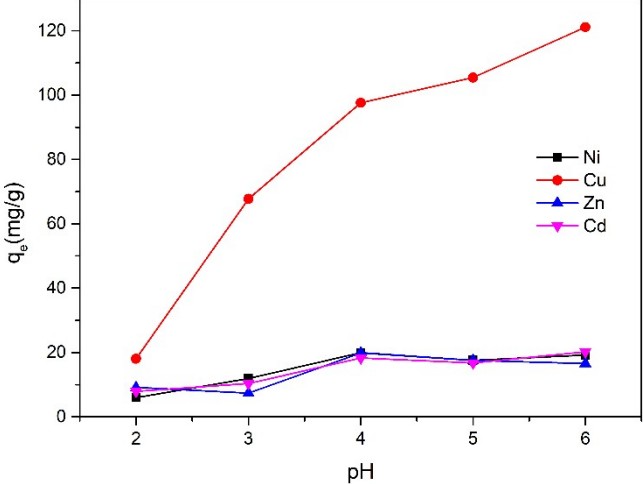

**Figure 6.** Effect of pH in a quaternary metal system. Reaction conditions: T = 293 K, $[Cu(II)]_0$ = $[Zn(II)]_0$ = $[Ni(II)]_0$ = $[Cd(II)]_0$ = 200 mg/L, reaction time = 12 h.

### 3.5. Adsorption Mechanism Analysis

FTIR was used to analyze the molecular structure of chemical compounds and the FTIR spectra of CA beads is shown before and after metal ion adsorption in Figure 7. For the CA beads before adsorption, the dominant peak at 3425.28 $cm^{-1}$ was due to the vibration stretching of the O-H bond, indicating that hydroxyl (-OH) groups existed in the beads. Bands at 2926.01 $cm^{-1}$ referred to the vibration stretching of -CH. The absorption peaks around 1610.24 $cm^{-1}$, 1418.32 $cm^{-1}$ and 1034.37 $cm^{-1}$ could be attributed to the asymmetric and symmetric stretching vibrations of -COO (carboxylate) and the stretching vibration of C-O, respectively [36]. After the sorption of metal ions, the FTIR spectra displacement slightly changed. This may be because the increased ionic volume weakened the stretching and torsional vibration of the functional groups, thus causing the displacement of adsorption peaks. The decrease of peak vibration intensity indicated that metal ions bind to -OH and -COO formed in the CA beads. In addition, no new peaks were produced, indicating that the functional groups of the adsorbents did not change, and the ion exchange process between the metal ions and the CA beads was possible.

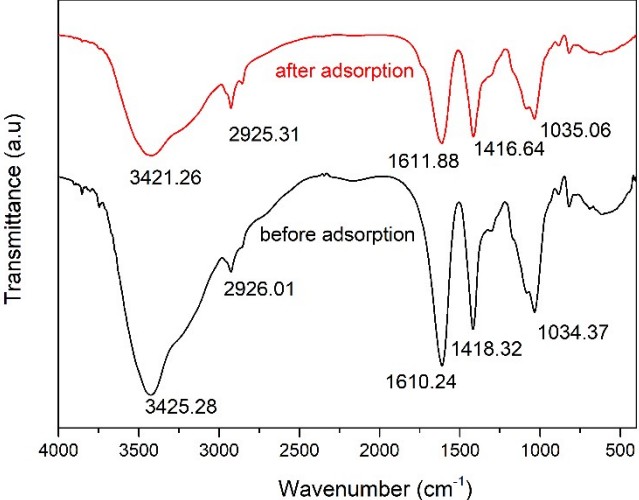

**Figure 7.** FTIR images of CA beads before and after the adsorption.

Figure 8 shows the XPS diagram of Cu 2p, Cd 3d, Zn 2p and Ni 2p (after sorption), further exploring the adsorption mechanism. The binding energy of Cu 2p, Cd 3d, Zn 2p and Ni 2p were

933.55/953.44 eV, 405.59/412.34 eV, 1021.97/1044.94 eV and 856.96 eV, respectively, after deconvolution, indicating that the adsorbed metal ions were in chemical states without further oxidation or reduction and the valence states were not changed during the ion exchange process [37–40].

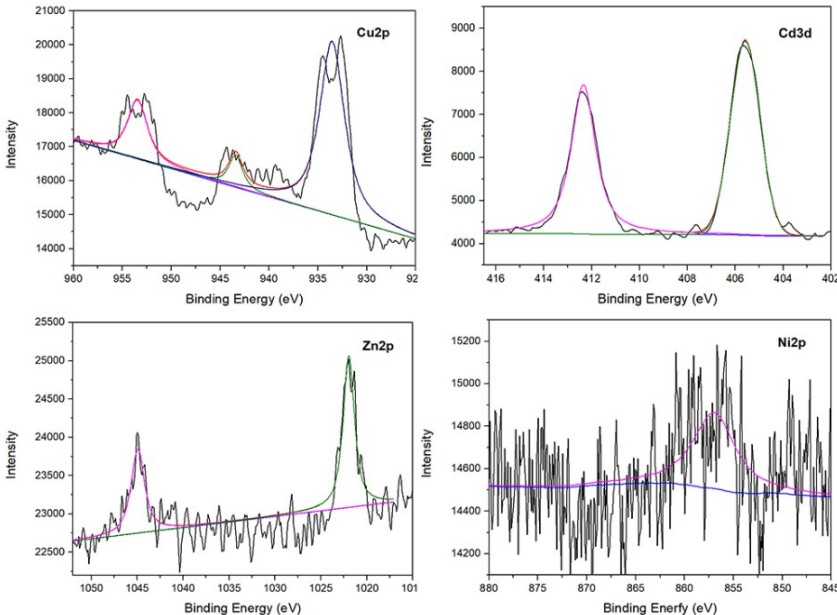

**Figure 8.** X-ray photoelectron spectroscopy (XPS) spectra of CA beads after the adsorption of mixed metal ions at a concentration of 500 mg/L in aqueous solution.

Cu(II) preferentially bound to the active sites of CA beads when the four ions were coexistent at the same concentration (Figure 9). The selective adsorption mechanism of copper ions on CA beads was via divalent metal ions exchanging ions with Ca(II) in the "egg box" structure [41]. The adsorption sites were mainly negative groups (COO-) that could adsorb the cation in the solution, and the stability of the bond with Cu(II) was better [42]. Therefore, compared with Zn(II), Ni(II) and Cd(II), the affinity for Cu(II) was the strongest in the ion exchange process between Ca(II) and metal ions in CA beads. This competitive relationship made the CA bead a potential material that could be applied to the recovery the copper ions from heavy metal mixtures.

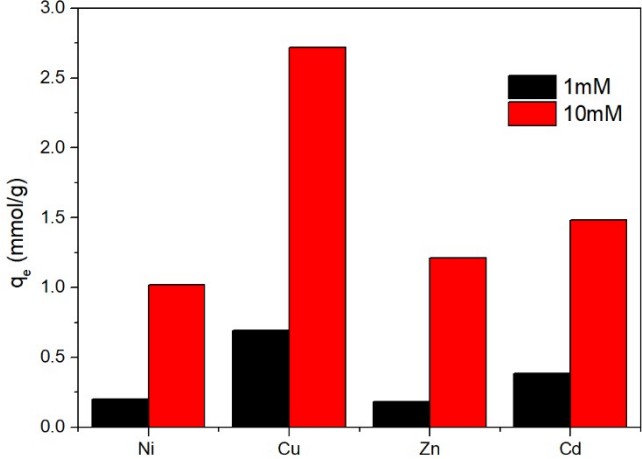

**Figure 9.** Effect of Cu(II) competitive adsorption in a quaternary metal system. Reaction conditions: T = 293 K, pH = 5, $[Cu(II)]_0 = [Zn(II)]_0 = [Ni(II)]_0 = [Cd(II)]_0 = 1$ mM/10 mM, reaction time = 12 h.

*3.6. Desorption and Reusability Experiment*

Regeneration of loaded sorbent is a key factor in water treatment processes and the desorption of the adsorbed Cu(II), Zn(II), Ni(II) and Cd(II) ions with 0.1 M HNO₃ was investigated. Figure 10 showed the reusability of CA beads with four heavy metal ions. The results revealed that after five cycles, the CA beads still had selective adsorption for Cu(II) and the adsorption capacity was more than 92%. Therefore, it can be explained that CA beads have good reusability during reuse experiments and have economic potential in wastewater treatment.

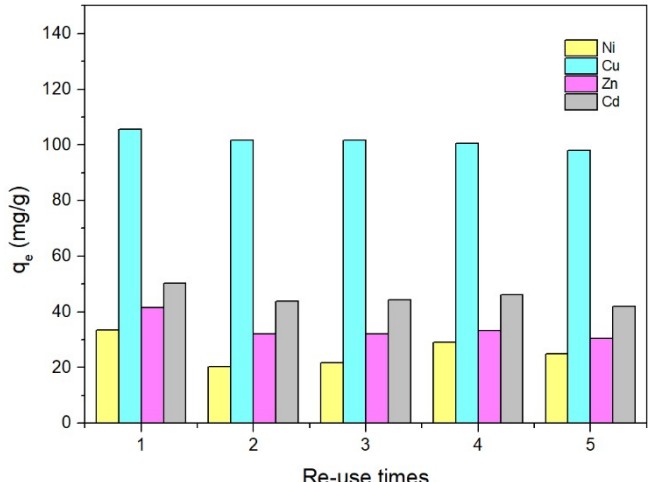

**Figure 10.** Effect of recycling on Cu(II), Zn(II), Ni(II) and Cd(II) ion adsorption. Reaction conditions: T = 293 K, $[Zn(II)]_0 = [Ni(II)]_0 = [Cd(II)]_0 = 200$ mg/L; reaction time = 12 h.

## 4. Conclusions

A spherical CA bead with a diameter of 3 mm was prepared by crosslinking the hydroxyl and carboxyl groups of sodium alginate with Ca(II) to form an insoluble hydrogel; there were a large number of active sites in the porous honeycomb structure for metal ions to attach. According to the pseudo-second-order and Langmuir isotherm model, the adsorption mechanism was explained well and the chemical reaction dominated the rate of the adsorption process. The maximum uptake of Cu(II) could reach 107.53 mg/g in the mixed heavy metal solution and the valence state of the four metal ions was not changed according to XPS analysis during the adsorption process. Cu(II) exchanged ions with Ca(II), binding with α-L-guluronic acid in the "egg box" structure. The selective adsorption was indicated through the isotherm experiments, giving this material a high potential for the continuous treatment of the selective recovery of copper from multi-metal solutions.

**Author Contributions:** Conceptualization, R.W. and P.R.; methodology, N.Y. and F.C.; software, L.Y. investigation, N.Y., R.W. and W.Z.; resources, R.W. and J.W.; writing—original draft preparation, N.Y.; writing—review and editing, N.Y., R.W. and P.R.; supervision, R.W. and P.R.; project administration, R.W., J.W. and P.R.; funding acquisition, R.W. and J.W.

**Funding:** This research was funded by Shanghai Sailing Program (Grant No. 17YF1407200), the "Capacity Building Project of Some Local Colleges and Universities in Shanghai" (Grant No. 17030501200), SUES Sino-foreign cooperative innovation center for city soil ecological technology integration (Grant No. 2017PT03) and the Project of Shanghai University Young Teacher Training Scheme (Grant No. ZZGCD16018).

**Conflicts of Interest:** The authors declare no conflict of interest.

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
