# Peer review of "The Fabrication of Calcium Alginate Beads as a Green Sorbent for Selective Recovery of Cu(Ⅱ) from Metal Mixtures"

_crystals, doi:10.3390/cryst9050255_

Round 1

Author Response

Responds to the reviewer’s comments:

Comment 1: Line 2 : Simple fabrication of alginate beadsThe title of the manuscript creates the expectation that a method of making beads of alginate was created or substantially modified. This is not true.

Response: Thanks to the referee’s comment, the title of the manuscript has been corrected in Line 2, and the key words of “Simple fabrication” was replaced by “Green sorbent” in line 28.

Comment 2: Line 15: “Abstract: Sodium alginate (SA) beads” Sodium alginate beads cannot be made because sodium alginate is soluble in water. Sodium alginate is used in the synthesis of calcium alginate beads. The manuscript should use the term calcium alginate beads instead of sodium alginate beads. The manuscript should include a brief description of the mechanism of formation of the calcium alginate beads.

Response: Thank you for underlining this deficiency. We have used the term calcium alginate beads instead of sodium alginate beads in this manuscript, and the mechanism of formation of the calcium alginate beads was added in Line 16-17, and the detailed introduction to the mechanism was added in Line 54-58.

Comment 3: Line 17 to 19: “The characterization and sorption behavior of the SA beads were analyzed in detail via using Scanning Electron Microscopy (SEM), Fourier Transform Infrared Spectroscopy (FTIR) and X-ray Photoelectron Spectroscopy (XPS).

They are characterization and qualitative analyses but a description of the methodology and equipment used for the quantitative analysis of the concentration of heavy metals in water is only described in one sentence in the lines 91 to 92: “The concentration of Cu(II), Zn(II), Ni(II) and Cd(II) in all experiments was determined by Inductively Coupled Plasma (ICP-OES, Varian 700-ES)”. A brief description of the procedure used to prepare the samples for the analysis by ICP-OES should be included, were the samples acidified?

Response: Thanks for the referee’s kind suggestion, we have added the procedure used to prepare the samples for the analysis by ICP-OES in Line 18 and Line 92-93.

Comment 4: Line 58 to 61: “Consequently, sodium alginate was a promising biomaterial for the biosorption of heavy metals [16]. In previous studies, it was found that sodium alginate had a selective adsorption effect on some metal ions. Hence, we want to investigate whether sodium alginate has selective adsorption on copper under the interference of cadmium, zinc and nickel metal ions.”

Sodium alginate is a powder soluble in water. The reference #16 refers to calcium alginate. The graphical abstract of the reference #16 is shown below.

Response: Thank you for underlining this deficiency. We have used the calcium alginate instead of sodium alginate in Lines 60-62.

Comment 5: Lines 80 to 88: “2.2. Preparation of alginate beads”.

The paragraph describes the usual procedure to prepare calcium alginate beads.

Response: Thanks for the referee’s kind suggestion. We did use the most commonly used method to synthesize calcium alginate beads, which was simpler and more economical than other methods.

Comment 6: Lines 105 to 106: “The sorption experiments were performed at room temperature with magnetic stirring and pH value uncontrolled.”

This means that each metal solution had its own pH level. The initial pH of each solution is not

mentioned. Was the initial pH measured? Could this affect the results of adsorption capacity and kinetics? Is valid to compare the adsorption capacity and kinetics between metals in solutions with different pH levels? The initial pH of each solution should be mentioned.

Response: Thank you very much for your question. The initial pH of Cu(II), Zn(II), Ni(II) and Cd(II) solutions are 5.5, 6.3, 6.6, 6.6, respectively, which were added in Lines 107-108. In this paper, we do have some deficiencies in controlling pH, and we will consider more comprehensive in future work.

Comment 7: Lines 125 to 126: “50 mL of pH was adjusted solution and 1 g calcium alginate beads were used in experiment.

The sentence makes no sense.

Response: Thanks for the referee’s kind suggestion, the sentence has been modified in Lines 128-130.

Comment 8: Lines 129 to 132: “Desorption experiments were also conducted using 1% CaCl2 solutions and 0.1 M HNO3 as the desorption reagents. After sorption, the heavy metals loaded SA beads was subsequently suspended in 0.1 M HNO3 eluting agents to evaluate the desorption performance and 1% CaCl2 solution was used to adjust the pH at 7.0”

Why the pH needs to be adjusted to pH 7 during the desorption process? How a solution of 0.1 M HNO3 could be adjusted to pH 7 using 1% CaCl2? The problem with that statement is that Cu(II) will precipitate in copper hydroxide at this pH and these concentrations of ppm. This procedure is not clear.

Response: Thank you very much for your question. The CA beads desorbed with HNO3 was acidic, so the CA beads were washed with CaCl2 to make it neutral and further solidified as the initial state. We had also made Changes in line 135-137.

Comment 9: Line 144: “Figure 1. SEM images of the outer surface (a-c) and inside (d) of the dried SA bead.”

What is the meaning of inside of the dried SA bead? Was the bead broken to take the SEM image?

Response: Thanks for the referee’s kind suggestion. After drying, the CA bead was broken, and the internal structure of the bead was observed by SEM. The “inside” was replaced by “the internal structure” in line 148.

Comment 10: Lines 153 to 157: “The point of zero charge (PZC) was employed to analyze the surface charges of SA beads. As shown in Figure3, the point of zero charge (PZC) was 8.2. When pH was lower than 8.2, the surface of the material was positively charged because of the introduction of calcium ions into the material. Above 8.2, there was a negative charge due to the reaction between the -OH on the surface and OH in the solution leading to form a negative charge functional group-O-.”

What is the point to include the results of this test? if the metals are soluble in an acidic environment below of pH level of 8.2. According to the results of this test, below of 8.2, the surface of the calcium alginate bead has a positive charge like the metal. These results would be useful with an explanation of why the metal ion with positive charge is attracted to a surface with positive charge.

Response: Thanks for the referee’s kind suggestion. After careful thinking, we also realized that the PZC had no meaning to the experimental results, so we decided to delete this paragraph.

Comment 11: Lines 175 to 176: “Figure 4. Kinetic adsorption plots of Cu(II), Zn(II), Ni(II) and Cd(II) by SA beads, respectively. Reaction conditions: T=293 K, uncontrolled pH, [Cu(II)]0=[Zn(II)]0=[Ni(II)]0=[Cd(II)]0=200 mg/L .”

Would be valid a comparison of results of kinetic of adsorption in solutions of different metals at different levels of pH?

Response: Thank you very much for your question. In this paper, we do have some deficiencies in controlling pH, and we will pay more attention to this problem in future work.

Comment 12: Lines 192 to 194: “Figure 5. The adsorption isotherm by SA beads in the Cu(II), Zn(II), Ni(II) and Cd(II) heavy metals solution at different initial concentrations. Reaction conditions: T=293 K, uncontrolled pH, [Cu(II)]0=[Zn(II)]0=[Ni(II)]0=[Cd(II)]0=0–800 mg/L, reaction time=12 h.”

Would be valid a comparison of results of capacity of adsorption in solutions of different metals at different levels of pH?

Response: Thank you very much for your question. The pH value difference of the Cu(II), Zn(II), Ni(II) and Cd(II) heavy metals solution with adjacent concentration is within 0.2, which results in little effect. Compared with the difference, the concentration is the main factor that affects the adsorption effect. In this paper, we do have some deficiencies in controlling pH, and we will consider more comprehensive in future work.

Comment 13: Line 216 to 219: “Figure 6. The selective removal and adsorption isotherm by SA beads in the mixed heavy metals solution at different initial concentrations. Reaction conditions: T=293 K, uncontrolled pH, [Cu(II)]0=[Zn(II)]0=[Ni(II)]0=[Cd(II)]0=0-800 mg/L, reaction time=12 h.”

The pH of this solution is unknown. The authors of this manuscript should affirm that the value of pH is not important in the adsorption of metals. This is not the case in this manuscript.

Response: Thank you very much for your question. The initial pH of 50, 100, 200, 300, 400, 600, 800 mixture solutions are 6.0, 5.8, 5.7, 5.6, 5.4, 5.3, 5.2, respectively. The pH value difference of the solution with adjacent concentration is within 0.2, which results in little effect. Compared with the difference, the concentration is the main factor that affects the adsorption effect. In this paper, we do have some deficiencies in controlling pH, and we will consider more comprehensive in future work.

Comment 14: Line 220 to 223: “3.4. Effect of pH in general, the pH of aqueous solution was an important role in the sorption process, while the degree of ionization, the surface charge and the speciation of metals could be affected by different pH values, resulting the different sorption mechanisms and the uptake capacities.”

This statement is a contradiction of the experimental methodology used in this manuscript.

Response: Thanks for the referee’s kind suggestion, we deleted this statement.

Comment 15: Line 229 to 231: “Figure 7. Effect of pH in a quaternary metal system. Reaction conditions: T=293 K, pH=2-6, [Cu(II)]0=[Zn(II)]0=[Ni(II)]0=[Cd(II)]0=200 mg/L, reaction time=12 h.”

The figures 6 and 7 deserve a comparison.

The figure 7 shows the adsorption in the range of pH from 2 to 6. The behavior of Cd, Zn and Ni in this figure is too similar, almost as if they were the same metal. This behavior could be possible. Nevertheless, the value of adsorption for Cadmium at 200 mg/L in the figure 6 is almost 40 mg/g but in the figure 7, the value of Cadmium at 200 mg/L in the range of pH from 2 to 6 never pass of 20 mg/g. The results are not consistent.

Response: Thank you very much for your question. We suspect that the cause of this phenomenon may be “In the pH experiment, the metal solution contains a amount of hydrogen ions, which separates the calcium ions of CA beads, and the exposed functional groups are limited to bind to copper ions, resulting in less redundant active sites to bind to other three metal ions.”.

Comment 16: Line 252 to 257: “To further explore the sorption mechanism, XPS studies of Cu2p, Cd3d, Zn2p and Ni2p (after sorption) were carried out and shown in Figure 9. The binding energy of Cu2p, Cd3d, Zn2p and Ni2p were 933.55/953.44 eV, 405.59/412.34 eV, 1021.97/1044.94 eV and 856.96 eV, respectively after the deconvolution indicating that the adsorbed metal ions were in chemical states without further oxidation or reduction and the valence states were not changed during the ion exchange process [27-31]”

How can the authors reach that conclusion? What would be the reference values of binding energy for these metals? The Nickel spectrum in Figure 9 is noise.

Response: Thank you very much for your question. We can know the binding energy parameters of these metals from the National Institute of Standards and Technology (NIST) and the BE Lookup Table for Signals from Elements and Common Chemical Species. the reference values of binding energy for Cu2p, Cd3d, Zn2p and Ni2p were 932.6/952.2 eV, 405.0/411.7 eV, 1021.81/1044.8 eV and 852.6 eV/869.9, respectively. Because of the magnetism of nickel, so that the spectrum being measured is noise.

Comment 17: Line 275 to 277: “Figure 10. Elemental mapping (C, N, O, Cu, Cd, Zn and Ni elements) after the adsorption in the mixed metal solution with the concentration of 50 mL/g (a-e) and 500 mL/g (fj).”

What it is the purpose of this figure? The legend of the colors is intelligible.

Response: Thanks for the referee’s kind suggestion. After careful thinking, we also realized that the PZC had no meaning to the experimental results, so we decided to delete this paragraph.

Comment 18: Line 289 to 291: “Figure 12. Effect of recycling on Cu(II), Zn(II), Ni(II) and Cd(II) ions adsorption. Reaction conditions: T=293 K, [Zn(II)]0=[Ni(II)]0=[Cd(II)]0=200 mg/L; reaction time=12 h.”

What is the pH level during each adsorption phase? The procedure used during the adsorption/desorption process is not clear.

Response: Thanks for the referee’s kind suggestion. The solution of heavy metal ions is the same and the initial pH value is about 5.7 in line 133-134. The adsorption/desorption process is described in line 135-137.

Comment 19: Finally, according to the authors: Why cadmium and zinc have a greater individual adsorption capacity than copper? Why in a mixture of metals, copper possesses a greater adsorption capacity? What is the difference? What is the reason of the inconsistency between the figure 6 and figure 7 with respect of cadmium? Why cadmium, zinc and nickel show a similar behavior in the figure 7 and copper has a totally different behavior?

Response: Special thanks to you for your good comments. The adsorption capacity of cadmium and zinc is greater than that of copper, which may be the fact that their relative molecular mass is greater than that of copper, especially cadmium.

We tried to use the DFT (Density Functional Theory) calculation to explain why the CA sphere has selective adsorption of copper ions, but this model had not been successfully established.

In the experiment of the figure 6, we did not adjust the pH value, and the initial pH value was about 5.7. In the experiment of the figure 7, we added HNO3 and NaOH to adjust the pH value. It may be that the regulation of pH causes the result to be different. Therefore, we suspect that the cause of this phenomenon is “In the pH experiment, the metal solution contains a large amount of hydrogen ions, which separates the calcium ions of CA spheres, and the exposed functional groups are limited to bind to copper ions, resulting in no redundant active sites to bind to other three metal ions.”, so cadmium, zinc and nickel show a similar behavior in the figure 7 and copper has a totally different behavior.

Reviewer 2 Report

This manuscript describes the use of calcium alginate as a sorbent for heavy metals. Authors focused on the selectivity of calcium alginate for copper among mixed heavy metal solution. This finding is very interesting and this special property can be useful for practical application. However, the high selectivity toward copper was well reported in previous literature. For example, Jodra and Mijangos (2012) already reported the high selectivity of calcium alginate gels for copper, and they used ion exchange equations to model the phenomena. This submitted article provides useful information regarding heavy metal adsorption, but I cannot find the originality. The regeneration study can be considered as a new finding, but I strongly suggest a continuous column study in order to emphasize its practical applicability.

In this regard, I do not recommend it to be published in Crystals in current form.

Jodra, Y., Mijangos, F. Ion exchange selectivities of calcium alginate gels for heavy metals. Water Sci Technol. 2001;43(2):237-44.

Author Response

Thank you for your letter and for the reviewers’ comments concerning our manuscript entitled “Simple fabrication of alginate beads as a green sorbent for selective recovery of Cu(II) from metal mixtures”. We have made in-depth revisions to the English of the whole article and corrected some mistakes. On the innovation of the article, we pay more attention to the application of adsorbents. The alginate beads can selectively recover Cu(II) ions from metal mixtures with high adsorption efficiency. After 5 cycles of adsorption and desorption, the uptake of Cu2+ on CA beads could reached more than 92% of the initial, which exhibited a good reusability potential. In addition, various characterization methods were used to analyze the morphology and structure of adsorbent, functional groups and adsorption mechanism of Cu(II).   

Once again, thank you very much for your comments.

We appreciate for Editors/Reviewers’ warm work earnestly and hope that the correction will meet with approval.

Reviewer 3 Report

Manuscript Ref.: Crystals-488488

Title: Simple fabrication of alginate beads as a green sorbent for selective recovery of Cu(II) from metal mixtures

This paper reports information and procedure on selective Copper(II) removal from aqueous solution by alginate beads. The study of removing of a metal ion from an aqueous solution  using alginate beads is not very innovative, but authors include experimental information and data obtained using different instrumental techniques and in different conditions of the system such as dependence on pH values.

However, the paper may be accepted after major revision.

Comments and specific suggestions:

The paper can be accepted for publication in the Crystals journal after the following suggestions and comments have been taken into account:

·         English throughout the text is to be reviewed in depth. Some sentences are unclear and seem meaningless.

·         Pag.1 – Introduction: I think that the recovery of copper it is not so important in the field of water treatment and in any case, it is not of primary importance; line 50-53: it’s not clear: Alginate and its derivatives have been well known and widely used in scientific research for many years

·         Pag.2 – 2.1 Materials: authors used 0.1 and 1M hydrochloric acid solution or HCl 36%? 

·         Pag.3 - Adsortpion isotherm: equation (2) is not referred to “maximum adsorption” but the adsorption of metal ion by the adsorbent material at equilibrium (qe).

·         Pag.3 - pH experiments. Lines 125-126 the sentence is incomprehensible (... pH meter. 50 ml of pH was...)

·         Pag.3 – Desorption and reuse experiments. Lines 131-132 the sentence is incomprehensible: have the authors used a solution of CaCl2 to adjust the pH?

·         Pag.3 – Desorption and reuse experiments. Do the alginate beads not disintegrate in a solution of 0.1M HNO3? The pH value is very low: the binding functional groups should be carboxyl groups and the egg-box type structure of alginate beads present a Ca2+ ion as “cross link agent” between different chemical groups.

·         Pag.5 – Adsorption kinetics. Adsorption kinetic studies are important for industrial process design but are also important because they allow us to know, under certain conditions of the system to be investigated, when the adsorption equilibrium is reached.

·         Pag. 6 – Figure 4. Why did the authors study the adsorption kinetics of different metal ions in different time intervals (until 1400min for Cu and Cd and until 550min for Zn and Ni)?

·         Pag.7 – Table 2. If it is necessary to compare the affinity or adsorption capacity of an adsorbent material against different metal ions, it is definitely better to use qe in mmol g-1 and not mg g-1.

·         For the adsorption isotherm experiment the authors prepared solutions of each metal up to 800 mg L-1. For mixed solution, with four metal ions, they reach 3,2 g L-1 of total metal ions and, perhaps, at this concentration the pH value can change also due to metal hydrolysis. For this reason, even if the pH is uncontrolled, it is important to know the initial pH value of the solution.

·         Pag.8 – Effect of pH. The method by which the points of Figure 7 were obtained is not clear. Is each point a qmax, a qe at a given value of pH or are they points obtained using a single solution in which the pH has been changed by adding a basic or an acidic solution?

·         Pag.9 - Figure 8. shows the images of the IR spectra relative to the beads before and after contact with the metal solution. Do the authors believe that we can really highlight a difference in response from the two spectra shown?

·         Pag 10 – Figure 11. q in lower case

A comparison could be made with literature data also on similar adsorbent materials.

Author Response

Comment 1: English throughout the text is to be reviewed in depth. Some sentences are unclear and seem meaningless.

Response: Thanks for the referee’s kind suggestion, the English of the article has been revised, that are marked in gray in the paper.

Comment 2: Pag.1 – Introduction: I think that the recovery of copper it is not so important in the field of water treatment and in any case, it is not of primary importance; line 50-53: it’s not clear: Alginate and its derivatives have been well known and widely used in scientific research for many years.

Response: Thanks for the referee’s kind suggestion, the description of “recovery of copper” has been modified in line 37-41. The line 50-53 has been modified in Line 51-52.

Comment 3: Pag.2 – 2.1 Materials: authors used 0.1 and 1M hydrochloric acid solution or HCl 36%?

Response: Thank you for underlining this deficiency, this error has been modified, the words of “0.1 and, 36%” were deleted in line 78.

Comment 4: Pag.3 - Adsortpion isotherm: equation (2) is not referred to “maximum adsorption” but the adsorption of metal ion by the adsorbent material at equilibrium (qe).

Response: Thanks for the referee’s kind suggestion, this error has been modified in line 121.

Comment 5: Pag.3 - pH experiments. Lines 125-126 the sentence is incomprehensible (... pH meter. 50 ml of pH was...)

Response: Thanks for the referee’s kind suggestion, the sentence has been modified in Lines 128-130.

Comment 6: Pag.3 – Desorption and reuse experiments. Lines 131-132 the sentence is incomprehensible: have the authors used a solution of CaCl2 to adjust the pH?

Response: Thanks for the referee’s kind suggestion, The CA beads desorbed with HNO3 was acidic, so the CA beads were washed with CaCl2 to make it neutral and further solidified as the initial state. We had also made Changes in line 135-137.

Comment 7: Pag.3 – Desorption and reuse experiments. Do the alginate beads not disintegrate in a solution of 0.1M HNO3? The pH value is very low: the binding functional groups should be carboxyl groups and the egg-box type structure of alginate beads present a Ca2+ ion as cross link agent” between different chemical groups.

Response: Thanks for the referee’s kind suggestion. Because of the α-L-guluronic acid (G) and β-D-mannuronic acid (M) of alginate will make the alginate beads more tightness in acidic conditions, so the alginate beads don’t disintegrate in a solution of 0.1M HNO3.

Comment 8: Pag.5 – Adsorption kinetics. Adsorption kinetic studies are important for industrial process design but are also important because they allow us to know, under certain conditions of the system to be investigated, when the adsorption equilibrium is reached.

Response: Thanks for the referee’s kind suggestion. This part has been added in line 159-160.

Comment 9: Pag. 6 – Figure 4. Why did the authors study the adsorption kinetics of different metal ions in different time intervals (until 1400min for Cu and Cd and until 550min for Zn and Ni)?

Response: Thank you very much for your question. The time interval of these four metals was 1440 minutes. Since the adsorption capacity of Zn and Ni did not change after 120 minutes, the adsorption equilibrium was basically reached, so we chose the time interval to 540 minutes.

Comment 10: Pag.7 – Table 2. If it is necessary to compare the affinity or adsorption capacity of an adsorbent material against different metal ions, it is definitely better to use qe in mmol g-1 and not mg g-1.

Response: Thanks for the referee’s kind suggestion. In Table 2, we want to highlight that in the mixture solution, the adsorption capacity of CA to copper ions is only slightly decreased, while the other three metal ions are much less than the amount of adsorption alone. I think it is more intuitive to use qe in mg g-1 than mmol g-1. What’s more, in Figure 11, we used qe in mmol g-1 to demonstrate the affinity.

Comment 11: For the adsorption isotherm experiment the authors prepared solutions of each metal up to 800 mg L-1. For mixed solution, with four metal ions, they reach 3,2 g L-1 of total metal ions and, perhaps, at this concentration the pH value can change also due to metal hydrolysis. For this reason, even if the pH is uncontrolled, it is important to know the initial pH value of the solution.

Response: Thank you very much for your question. The initial pH of 50, 100, 200, 300, 400, 600, 800 mixture solutions are 6.0, 5.8, 5.7, 5.6, 5.4, 5.3, 5.2, respectively. The pH value difference of the solution with adjacent concentration is within 0.2, which results in little effect. Compared with the difference, the concentration is the main factor that affects the adsorption effect. In this paper, we do have some deficiencies in controlling pH, and we will consider more comprehensive in future work.

Comment 12: Pag.8 – Effect of pH. The method by which the points of Figure 7 were obtained is not clear. Is each point a qmax, a qe at a given value of pH or are they points obtained using a single solution in which the pH has been changed by adding a basic or an acidic solution?

Response: Thanks for the referee’s kind suggestion, each point is a qe, the points obtained a series of solutions that the pH has been changed by adding a basic or an acidic solution. We have made some modifications in this manuscript. See Line 128-130 for details.

Comment 13: Pag.9 - Figure 8. shows the images of the IR spectra relative to the beads before and after contact with the metal solution. Do the authors believe that we can really highlight a difference in response from the two spectra shown?

Response: Thanks for the referee’s kind suggestion. Because there is no change in the number of infrared functional groups before and after adsorption, we can know that there is no oxidation or reduction reaction in the adsorption process. We have made some modifications in Line 243-249.

Comment 14: Pag 10 – Figure 11. q in lower case

Response: Thanks for the referee’s kind suggestion, this error has been modified in Figure 9.

Comment 15: A comparison could be made with literature data also on similar adsorbent materials.

Response: Thanks for the referee’s kind suggestion. A comparison had been made on similar adsorbent materials in Line 187-189 and table 3.

Round 2

Reviewer 2 Report

This manuscript describes the use of calcium alginate beads for copper removal from metal mixtures. The selective feature of calcium alginate beads toward copper is very interesting and it is potentially promising to be a suitable technology to recover copper from the wastewater stream. The quality of the manuscript was significantly improved through a revision round. I found a few minor errors required to be revised before acceptance. I would like to recommend it to be published with minor revision.

I strongly suggest providing more intensive literature reviews regarding the use of calcium alginate beads for heavy metal adsorption. I am quite sure that there is a number of similar studies using calcium alginate. With an intensive literature review, it will be easier to see what the research novelty is.

Section 2.4 – 2.6: I suggest to re-organize these sections. I could see that overall experimental procedures are similar in these sections. Authors may use subsections in order not to write the same sentences in each section.

Section 3.4: Authors claimed that the surface charge has a dominant role on the effect of pH. I suggest providing point-of-zero-charge of CA bead in order to have a better understanding.

Page 283 (figure 9): I can see that figure 9 is the almost the same result with figure 5. And the overall discussion is also quite similar to section 3.3. I suggest re-organizing the result & discussion section.

Page 283 Line 263-267: Authors claimed that the adsorption of copper ions was increased significantly with the increase of the metal ion concentration compared to other ions. However, I think it is not just a matter of concentration, it will be more related to the ratio. It seems that the increase of adsorption capacity in increased concentration was around 4 times compared to 1 mM. It is quite similar to the feature of affinity in ion exchange resin. I suggest adding more discussion regarding interpreting the figure 9 with the affinity term in ion exchange.  

Author Response

Responds to the reviewers comments:

Comment 1: Section 2.4 – 2.6: I suggest to re-organize these sections. I could see that overall experimental procedures are similar in these sections. Authors may use subsections in order not to write the same sentences in each section.

Response: Thanks for the referee’s kind suggestion, these sections have been revised in Line 106 - 121.

Comment 2: Section 3.4: Authors claimed that the surface charge has a dominant role on the effect of pH. I suggest providing point-of-zero-charge of CA bead in order to have a better understanding.

Response: Thanks for the referee’s kind suggestion, the point-of-zero-charge of CA bead has been added in Line 223 - 227.

Comment 3: Page 283 (figure 9): I can see that figure 9 is the almost the same result with figure 5. And the overall discussion is also quite similar to section 3.3. I suggest re-organizing the result & discussion section.

Response: Thank you for underlining this deficiency. Compared with figure 5, we chose the same molar concentration in figure 9 to avoid the influence of the relative molar mass on the experimental results, and further illustrated the CA beads did have a strong affinity for copper ions.this section has been revised in Line 257 - 264.

Comment 4: Page 283 Line 263-267: Authors claimed that the adsorption of copper ions was increased significantly with the increase of the metal ion concentration compared to other ions. However, I think it is not just a matter of concentration, it will be more related to the ratio. It seems that the increase of adsorption capacity in increased concentration was around 4 times compared to 1 mM. It is quite similar to the feature of affinity in ion exchange resin. I suggest adding more discussion regarding interpreting the figure 9 with the affinity term in ion exchange. 

Response: Thanks for the referee’s kind suggestion, this section has been revised in Line 257 - 264.

Reviewer 3 Report

I would recommend to cite the article here reported concerned the metal removal from aqueous solution in different condition of the system by low-cost adsorbent:

S. Cataldo, A. Gianguzza, A. Pettignano, D. Piazzese, S. Sammartano, Complex formation of copper(II) and cadmium(II) with pectin and polygalacturonic acid in aqueous solution. An ISE-H+ and ISE-Me2+ electrochemical study, Int. J. Electrochem. Sci. 7 (2012) 6722–6737.

Author Response

Comment: The paper can be accepted for publication in the Crystals journal after the following suggestions and comments have been taken into account:

·I would recommend to cite the article here reported concerned the metal removal from aqueous solution in different condition of the system by low-cost adsorbent:

S. Cataldo, A. Gianguzza, A. Pettignano, D. Piazzese, S. Sammartano, Complex formation of copper(II) and cadmium(II) with pectin and polygalacturonic acid in aqueous solution. An ISE-H+ and ISE-Me2+ electrochemical study, Int. J. Electrochem. Sci. 7 (2012) 67226737.

Response: Thanks for the referee’s kind suggestion, this article has helped us a lot and has been cited in line 260 and added in the reference (36).
